# Valorization of waste pharmaceutical residues via pyrolysis: Simultaneous production of biochar for Cd²⁺ removal and high-quality bio-oil/syngas

Zhizhen Feng[1,2,3], Bo Fu[1,2,3], Shanshan Han[1,2,3], Hong Yan[1,2,3], Puyang Feng[1,2,3], Wenxiao Li[1,2,3], Tao Qin[1,2,3], Tongtong Wang[4], Xinjie Zhang[5], Junchao Jia [1,2,3]*

1 Shaanxi Key Laboratory of Qinling Ecological Security, Bio-Agriculture Institute of Shaanxi, Shaanxi Academy of Sciences, Xi'an, Shaanxi, P.R. China, 2 Shaanxi Key Laboratory of Plant Nematology, Xi'an, Shaanxi, P.R. China, 3 Enzyme Engineering Research Center of Shaanxi Province, Xi'an, Shaanxi, P.R. China, 4 Institute of Interdisciplinary and Innovation Research, Xi'an University of Architecture and Technology, Xi'an, Shaanxi, P.R. China, 5 School of Chemical and Environment Science, Shaanxi University of Technology, Hanzhong, Shaanxi, P.R. China

* jiajc@xab.ac.c

## Abstract

For the efficient utilization of pharmaceutical waste resources, tuber biochar (TB) and herbal biochar (HB) were prepared via oxygen-limited slow pyrolysis at 500 °C for 3 h, using residues from tuber-type Xinsuning capsule and herbal-type Changyanning pill as the raw materials, respectively. The biochars were characterized by FESEM, BET, XRD and FTIR, and the feedstock physico-chemical properties were measured by common agricultural chemical analysis methods. The results revealed that both biochars possessed a high percentage of elemental O, a honeycomb-like porous structure, and surfaces enriched with functional groups such as hydroxyl, carboxyl, and carbonyl. HB exhibited a larger specific surface area and pore volume than TB, making it a more recommended carbon material. The chemical compositions of the pyrolysis by-products were systematically analyzed. The bio-oils were rich in ketones, alkanes, alcohols, olefins, fatty acids, phenols, and heterocyclic compounds, identifying them as potential sources of liquid fuels and chemical feedstocks. The most abundant components in bio-oils from tuber and herbal biomass were "Ethanol, 2,2-diethoxy-" (7.25%) and "Phosphonic acid, (p-hydroxyphenyl)-" (10.52%), respectively. The syngas has a low hydrogen content, is mainly pyrolysis off-gas and therefore has a limited application potential. Furthermore, the environmental application for Cd²⁺ removal was critically evaluated. Adsorption isotherms demonstrated high adsorption capacities, well-described by the Freundlich model ($R^2 \geq 0.99$), indicating multilayer adsorption. The maximum adsorption capacities for TB and HB were 188.89 and 186.67 mg·g⁻¹, respectively. Kinetic studies revealed that the adsorption process followed the Elovich model ($R^2 \geq 0.98$), suggesting heterogeneous diffusion, with HB achieving a higher equilibrium capacity (85.67 mg·g⁻¹) than TB (73.70 mg·g⁻¹). In conclusion, pyrolysis, particularly using herbal biomass, presents a

**Data availability statement:** All relevant data are within the manuscript.

**Funding:** This research was supported by the Science and Technology Program of Shaanxi Academy of Sciences (Grant No.: 2024k-02), the Science and Technology Program of Xianyang (Grant No.: L2022-QCYZX-NY-006), the Shaanxi Provincial Key R&D Program (Grant No.: 2023-ZDLNY-54), and the Shaanxi Provincial Innovation Capacity Support Program (Grant No.: 2023-CX-PT-12).

**Competing interests:** The authors have declared that no competing interests exist.

promising strategy for the comprehensive and high-value utilization of waste pharmaceutical residues, simultaneously producing effective adsorbents for heavy metal remediation and valuable bio-energy products.

## Introduction

China is a large country of traditional Chinese medicine (TCM), and in recent years, the TCM industry has been developing rapidly, especially with Tu Youyou winning the Nobel Prize in Medicine for extracting artemisinin, TCM has been increasingly recognized in the world [1]. Currently, the planting area of Chinese herbs in China has reached more than 2.4 million hectares, and the annual output of herbs is about 70 million tons [2]. However, a large amount of waste pharmaceutical residue is subsequently generated during the processing of waste pharmaceutical tablets and the production of waste pharmaceutical decoction [3]. While the Chinese medicine industry is booming, the problem of waste pharmaceutical residue disposal is becoming increasingly serious. According to incomplete statistics, there are more than 1,500 Chinese medicine enterprises in China, and the annual emission of waste pharmaceutical residue can be as high as 60–70 million tons [4]. waste pharmaceutical residues are generally wet materials that have the characteristics of high organic matter, high moisture and perishable, strong odor, and so on [5]. At present, the treatment of waste pharmaceutical residue mainly adopts incineration, landfill and piling, and these traditional treatment methods not only cause a huge waste of resources, but also cause potential environmental pollution, such as soil, air, and water pollution, which finally endangers human health and impedes the green and sustainable development of the Chinese medicine resource industry [6]. With the strengthening of China's environmental governance, how to carry out the harmless treatment and resource utilization of waste pharmaceutical residue has become an important issue facing the Chinese medicine industry. Therefore, there is an urgent need to find effective high-value utilization methods for waste pharmaceutical residue.

The waste pharmaceutical residues contain a large number of crude fibers, starch, crude fat, protein, amino acids, and trace elements [7]. The bioconversion technology and thermochemical conversion technology are used to transform waste pharmaceutical residues into resource substances with high utilization value, so as to increase the added value of products and fully exploit the resource potential of Chinese medicine wastes [8]. At present, the resource utilization of waste pharmaceutical residue mainly involves planting, breeding, environment, energy, and many other fields. With the rapid development of the economy, environmental and energy issues are becoming more and more serious. As a kind of renewable organic solid waste, waste pharmaceutical residue, unlike fossil fuel, can be regarded as a carbon-neutral fuel with no net increase of carbon dioxide during use, which is expected to partially replace traditional fossil energy and alleviate the energy crisis [7]. "Carbon peak" and "Carbon neutrality" has become China's strategic goal, in the "Double carbon" background, the traditional Chinese medicine industry is ushering in a new wave of "Low

carbon" trends, and the high-value utilization of waste pharmaceutical residue has become a breakthrough for implementing a low-carbon economy in the Chinese medicine industry.

Numerous studies revealed that waste pharmaceutical residue can be converted into energy, energy carriers, and value-added chemicals through bioconversion (e.g., anaerobic digestion) and thermochemical conversion (e.g., direct combustion, pyrolytic carbonization, high-temperature liquefaction, and pyrolysis for gas production, [9–15]). (In this paper, "pyrolysis" is defined as oxygen-limited thermal decomposition, while "pyrolysis for gas production" refers to the process of producing $H_2$/CO-rich syngas via partial oxidation, steam reforming, or $CO_2$ reforming.) Shang et al. [11], Cheng et al. [12], and Lian et al. [13] used the pyrolytic carbonization technique to prepare waste pharmaceutical residue into biochar. Shang et al. [11] prepared biochar carriers from medicinal dregs of Scutellaria baicalensis Georgi and synthesized a nanoscale zero-valent iron particle composite for the adsorption of hexavalent chromium from wastewater. Cheng et al. [12] prepared biochar using medicinal dregs of Melicope pteleifolia at different temperatures to study its effect on the removal of tetracycline from aqueous solution. Lian et al. [13] used medicinal dregs Salvia miltiorrhiza Bunge to prepare biochar at different temperatures to study its adsorption capacity for the antibiotic sulfamethoxazole. Jiang et al. [14] and Zhang et al. [15] prepared medicine herb residue into bio-oil using high-temperature liquefaction technology. Jiang et al. [14] prepared liquid bio-oil by reducing the oxygen content in the process of rapid heating of medicine herb residue, which increased the added value of medicine herb residue. Zhang et al. [15] used ZSM-5 molecular sieves as catalysts to catalyze the rapid pyrolysis of medicine herb residue for the preparation of biofuel through microwave-induced methods. Ding et al. [16], Li et al. [17], and Chen [18] prepared medicine herb residue into biogas using pyrolysis for gas production. Ding et al. (2018) used CaO as a catalyst for the preparation of gas by pyrolysis for gas production of medicine herb residue, to study the catalytic property changes of the catalyst during the pyrolysis for gas production process as well as to investigate the mechanism of pyrolysis for gas production using thermodynamic simulation. Li et al. [17] carried out a pyrolysis for gas production experiment of medicine herb residue in a fixed-bed system using paper sludge rich in various metal oxides as an additive. The results showed that the medicine herb residue was catalyzed by paper sludge during pyrolysis, which led to an increase in gas production and a decrease in tar production. Chen et al. [18] carried out high temperature co-gasification experiment with medicine herb residue by adding printing and dyeing sludge. The results showed that the co-gasification of these two wastes could produce synergistic effects. The combined gasification performance of co-gasification was 33.9% and 33.2% higher than that of single pyrolysis of dye sludge under $N_2$ and $CO_2$ atmospheres, respectively. Compared with bioconversion (e.g., anaerobic digestion) and direct combustion, oxygen-limited pyrolysis offers advantages for pharmaceutical residues: it can simultaneously produce a stable solid carbonaceous adsorbent (biochar) and liquid products (bio-oil) that contain value-added chemicals, while reducing direct emissions associated with combustion [19]. For wet, high-organic residues typical of Chinese medicinal residues, pyrolysis at controlled temperatures allows recovery of both carbon materials and bioenergy precursors, making it a suitable route for high-value utilization [20].

Biomass pyrolysis is a high temperature anaerobic process with a temperature range of 300~1000°C [21]. Typically, biochar, bio-oil, and syngas were produced by pyrolysis of medicine herb residue [22]. Biochar could be processed into adsorbent, activated carbon, etc., and used in wastewater treatment, chemical industry, and smelting. Bio-oil could be converted into green chemicals. syngas could be directly converted into liquid fuel oil for power generation, supplying electricity to residents and enterprises [23]. Recent studies have highlighted that biochars' adsorption performance for heavy metals is controlled by multiple factors including surface functional groups (–OH, –COOH), specific surface area, pore size distribution, and the presence of mineral phases that facilitate ion exchange or precipitation [24–26]. In particular, studies by El-Nemr et al. [24,25] and Mohamed et al. [26] demonstrate improved $Cd^{2+}$ removal with biochars having abundant oxygen-containing groups and mesoporous structure. In this study, two types of biochar were prepared from medicine herb residues as raw material by setting up two treatments, tuber-type Xinsuning capsule (tuber) and herbal-type Changyanning pill (herb), using oxygen-limited pyrolysis in a nitrogen atmosphere. The basic physicochemical properties and microstructures of two biochars were characterized using advanced techniques such as field emission

scanning electron microscopy (FESEM), surface elemental analysis, BET surface area and pore size analysis (BET), X-ray diffraction (XRD), and Fourier transform infrared (FTIR). In addition, the chemical compositions of the pyrolysis by-products, bio-oil, and syngas, were systematically analyzed, and different biomass pyrolysis products were compared. Also, the adsorption performance of $Cd^{2+}$ by both biochar types, including isothermal adsorption and adsorption kinetics, will be thoroughly investigated. It is expected to provide a theoretical and practical basis for the effective utilization of medicine herb residue resources and the development of eco-friendly biochar products and valuable by-products.

## Materials and methods

### Materials

**Feedstock source.** All solvents and chemicals are analytically pure and purchased from China National Pharmaceutical Group Chemical Reagent Co. Ultrapure water will be used throughout the experiments. The raw materials (medicine herb residue) were collected from the medicine factory of Shaanxi Momeide Pharmaceutical Co., LTD., Xianyang, Shaanxi Province, China. Waste drug residues are wet residues generated during the production process of the pharmaceutical plant, which are crushed and sieved after natural air drying, without chemical pretreatment. The two kinds of Chinese medicine residues were the herbal-type Changyanning pill (herb) and tuber-type Xinsuning capsule (tuber) in which the main components of the Changyanning pill were Euphorbia maculata Willd., Hedyotis chrysotricha (Palib.) Merr., Litsea rubescensLeeomtet, Elsholtzia Ciliata (Thunb.) Hyland. and Aceracede leaves. The main ingredients of the Xinsuning capsule were Coptis chinensis Franch., Pinellia ternata (Thunb.) Smilax glabra Roxb., Citrus aurantium L., Dichroa febrifuga Lour., Nelumbo nucifera Gaertn., Sophora flavescens Alt., Artemisia carvifolia, Panax ginseng C. A. Mey, Ophiopogon japonicus (Linn. f.) Ker-Gawl. and Glycyrrhiza uralensis Fisch.

**Feedstock characterization.** The samples of Chinese medicine residues were naturally air-dried, ground, and passed through sieves of different pore sizes. The basic physicochemical properties were measured. According to the determination method of Lu et al. [27], organic carbon and organic matter were determined by the potassium dichromate volumetric method; pH value was determined by the potentiometric method; total nitrogen was determined by the Kjeldahl nitrogen determination method; total phosphorus was determined by the alkali fusion-molybdenum antimony anti-spectrophotometric method; total potassium was determined by the alkali fusion-flame photometric method; crude protein was determined by the Kjeldahl nitrogen determination method; cellulose and hemicellulose were determined by the Van Soest method.

### Preparation of biochar, bio-oil, and syngas via pyrolysis

Twenty grams of dried waste pharmaceutical residues were weighed and placed into a box-type high-temperature furnace with nitrogen protection, where pyrolysis was performed throughout the entire process. In this study, low-temperature slow pyrolysis was employed with a multi-stage program heating protocol: the heating rate was maintained at 5°C·min$^{-1}$, the pyrolysis temperature was set at 500°C with a 3 h isothermal holding period, and the system was allowed to cool naturally to room temperature after the completion of the reaction. The solid obtained was ground through an 18-mesh sieve, and two treatments of waste pharmaceutical residues from tubers and herbs were set up, i.e., tuber pharmaceutical residues biochar (TB) and herb pharmaceutical residues biochar (HB) were obtained. Subsequently, TB and HB were crushed into small pieces of approximately 2 cm by a pulveriser (Zhejiang Fengli Pulverization Equipment Co., Ltd., Shaoxing, China). The ultrasonic activation process was carried out in the KQ-300ES constant temperature numerical control ultrasonic machine (Kunshan Ultrasonic Instrument Co., Ltd., Kunshan, China). During the ultrasonic process, a sufficient amount of biochar was first weighed into a beaker, and 200 ml of deionized water was added for dispersion. Then, the common basic setting for ultrasonic time was 30 min, the frequency was 40 kHz, and the power was 300 W. Finally, all solutions were filtered through a 0.45 um filter membrane to recover biochar, then dried at 105 °C for more than 8 hours.

## Characterisation

**Biochar.** According to Zhang *et al.* [28], the surface morphology of the biochar samples was analyszd in a Sigma HD scanning electron microscope (Carl Zeiss AG, Oberkochen, Germany) at a 2.0 kV, which was loaded with an X-MAXEDS spectroscopic detection system (Oxford Instruments, Oxford, UK). The BET Specific surface area ($S_{BET}$) and pore size analysis were determined using $N_2$ as the adsorbate at 77 K and relative pressure of 0.05–0.20, for which an ASAP2460 plus analyzer (Micromeritics Instruments Co., Norcross, USA) was used. The mineral species of the biochar samples were identified using a D-MAX 2500 X-ray powder diffractometer (Rigaku Co., Ltd., Wilmington, USA). The measurements were performed at a 0.02 scan step size, 2 deg·min$^{-1}$ scan speed, 0.15 receiving slit width, 30−40 kV, and 30−40 mA. The infrared spectra were measured with KBr pellet methods using a Vertex70 FTIR spectrometer (Bruker Co., Ltd., Billerica, USA) for 16 scans over a range of 400～4000 cm$^{-1}$ with a resolution of 2 cm$^{-1}$.

**Bio-oil and syngas.** The gas chromatography-tandem mass spectrometer (GC-MS, Agilent 7890A/7000B GC-QQQ, Santa Clara, USA) was used to analyze the components of bio-oil and syngas. GC-MS analysis conditions: HP-5 (30 m×0. 25 mm×0. 25 m) column; the temperature of vaporization chamber is 260 °C; High purity helium as carrier gas, and flow rate 1.0 mL·min$^{-1}$. For shunt injection, the shunt ratio is 25:1; The column temperature is programmed: the initial temperature is 50 °C and kept for 5 min, the temperature is raised to 100 °C at 2 °C·min$^{-1}$, and then the temperature is raised to 180 °C at 1 °C·min$^{-1}$; and finally the temperature was raised to 260 °C at 1 °C·min$^{-1}$ and kept for 5 minutes. MS electron energy 70 eV, filament current 80 A, ion source temperature 230 °C.

## Adsorption tests

A standard Cd solution of known concentration was prepared with analytically pure $Cd(NO_3)_2 \cdot 4H_2O$. Add 50 mL of cadmium nitrate solution of known concentration and 0.05 g of biochar sample to each test tube respectively, and shake at room temperature (25 °C) at 150 r·min$^{-1}$. After oscillation, samples were taken for solid-liquid separation. The concentration of $Cd^{2+}$ in the filtrate will be determined by Thermo Scientific ICE 3300 atomic absorption spectrophotometer. The initial concentrations of $Cd^{2+}$ pollutants were set at 1, 5, 10, 30, 50, 100, 150, and 250 mg·L$^{-1}$. The adsorption equilibrium time was uniformly set at 24 h. For the adsorption kinetics at an initial concentration of 100 mg·L$^{-1}$, samples were collected at intervals of 5, 10, 30, 60, 120, 180, 300, 480, 720, 900, 1440, and 2160 min to determine equilibrium concentrations and adsorption capacities. Relevant calculation methods can be referenced in [23]. To better investigate the characteristics of the adsorption, four isotherms models (Langmuir, Freundlich, Temkin, and Dubinin-Radushkevich) and four adsorption kinetic models (Pseudo-first-order, Pseudo-second-order, Elovich, and Intraparticle diffusion) were used in this work [23,28].

## Data analysis

In this work, data statistics were calculated using Excel 2016 Pro (Microsoft, Redmond, USA) and SPSS 26.0 (IBM Corporation, Armonk, USA). One-way ANOVA was conducted to compare the means of the measured values at $P < 0.05$ for each treatment. The XRD was analyzed by MDI Jade v6.5 (Materials data Inc., Livermore, USA) to identify the crystal data. Identification of FTIR spectroscopy could be identified using the spectral database (https://rruff.info/) or the "Spectral Tools" website (http://www.science-and-fun.de/tools/). OriginPro 2019b (OriginLab, USA) was used to calculate the optical properties of prepared materials and make plots.

## Results and discussions

### Characterization

**Basic properties.** The basic physico-chemical properties of the two kinds of Chinese medicine residues are shown in Table 1. Overall, the basic physicochemical properties of tuber biomass and herbaceous biomass are very similar,

Table 1. Basic physico-chemical properties of the raw materials.

| Materials | Organic carbon | Organic matter | pH | Total nitrogen | Total phosphorus | Total potassium | Crude protein | Cellul-ose | Hemicellul-ose |
|---|---|---|---|---|---|---|---|---|---|
| | % | g/kg | 1:10 | % | % | % | % | % | % |
| **Tubers biomass** | 39.4 | 679.256 | 5.42 | 1.43 | 0.201 | 0.227 | 8.957 | 21.8 | 43.1 |
| **Herbal biomass** | 42.6 | 734.424 | 5.74 | 1.40 | 0.176 | 0.229 | 8.757 | 29.2 | 35.8 |

which is attributed to the fact that they are both derived from Chinese medicinal raw materials. Specifically, the organic carbon and organic matter contents of tuber biomass were significantly lower than those of herbal biomass. This may imply that herbaceous biomass has a significantly higher C elemental content after carbonization than tuberous biomass and may have superior properties as a carbon material than those prepared from tuberous biomass. Both tuber and herbaceous biomass are acidic, which is closely related to the organic acids contained in them [18]. The total nitrogen, total phosphorus, and total potassium contents of tuber and herbaceous biomass were very similar, and the total nitrogen content was significantly higher than the total phosphorus and total potassium, implying an abundance of cellulose and hemicellulose [23]. The crude protein and hemicellulose content of tuberous biomass is higher than that of herbaceous biomass, and its cellulose content is lower than that of herbaceous biomass. This will affect the carbonization process and consequently lead to differences in the properties of the prepared carbon materials.

**FESEM and surface elemental analysis.** The FESEM (Scanning electron microscope) images of the prepared materials are displayed in Fig 1. From Fig 1(a),(c), it could be seen that TB and HB have obvious honeycomb polyhedral structures, and uneven surfaces, abundant porous structures with particles, resulting in a squamous texture and chaotic arrangement. Fig 1(b),(d) show the biochar magnified 6,000 times, with flakes of tubular structure and broken particles of different sizes and shapes distributed on the surface. Additionally, a sponge-like tube wall could be observed, and the enriched pores are very important as they provide the specific surface area and the active sites. In general, the micromorphology of TB and HB is similar, which is attributed to their raw materials being essentially the same. However, a careful comparison still reveals that HB has thicker spongy and capillary fibers with greater surface roughness than TB, which may lead to greater pore volume and shape a more developed microporous structure [29].

The EDS spectra of the prepared materials are shown in Fig 2. It may be observed that both TB and HB surfaces contain the elements C, N, O, P, S, and K. Since they are carbon materials, the content of C element is obviously more than half, followed by O and N elements. Also, TB and HB contained some heterogeneous elements, which were related to the enriched elements of biomass in the pharmaceutical residue [30,31]. In addition, O/C has been used to determine the aromatic structure and cation exchange capacity (CEC) of the polymers, and a larger value of this ratio implies a higher aromaticity and a larger CEC [23]. It is observed that HB has a higher O elemental content and a relatively lower C elemental content, which may imply that HB has a stronger surface polarity than TB, with a greater abundance of oxygen-containing functional groups such as hydroxyl, carboxyl and carbonyl groups. N/C may reflect the degree of decomposition of biomass during pyrolysis, and it can be found that the N/C value of TB is greater and the pyrolysis process is more intense [28], due to the intensification of dehydrogenation and deoxygenation reactions during pyrolysis, which is related to the fact that the feedstock of TB contains more lignin and cellulose (Table 1).

**XRD analysis.** The XRD patterns (Fig 3) revealed that the principal diffraction peaks at low $2\theta$ angles were sharp and symmetric, indicating the presence of mineral crystals. Overall, Fig 3 shows that TB and HB have similar XRD patterns, however, the peak patterns of HB are more pronounced than those of TB, and the mineral is more crystallized. Although there is a lot of noise signal interference in the XRD of the prepared materials in this work, it is roughly seen that there are two characteristic peaks at 24.4°, 26.4° and 29.4°, which were found to indicate the presence of impurities (may be $CaCO_3$, KCl or $SiO_2$) and graphitic carbon (JCPDS No.41–1487) respectively after searching for Jade 6.5 PDF cards. The

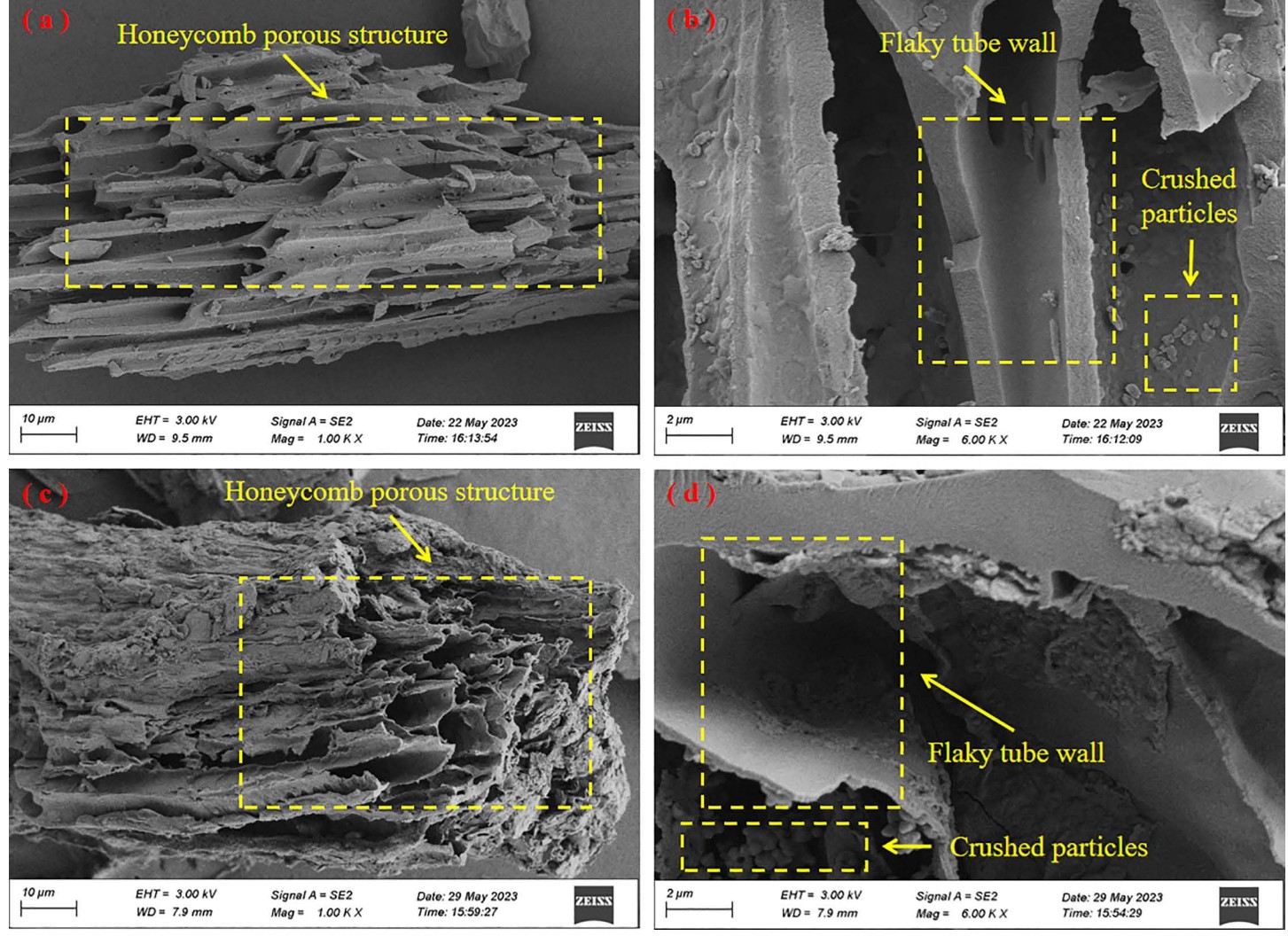

**Fig 1. FESEM images of the prepared materials. (a)** TB, 1000x magnification; **(b)** TB, 6000x magnification; **(c)** HB, 1000x magnification; **(d)** HB, 6000x magnification.

impurities are mainly cellulose and lignin pyrolysis process and release a large amount of ash, which contains some Ca, K, and other elements [23, 28]. The presence of mineral phases such as $CaCO_3$, KCl or $SiO_2$ modulates $Cd^{2+}$ adsorption in different ways [32]. Specifically, alkaline oxides such as $CaCO_3$ can raise the local pH, providing basic sites and favoring $Cd(OH)_2$ precipitation and/or inner-sphere surface complexation, which often enhances $Cd^{2+}$ removal [33]. Soluble salts like KCl increase ionic strength and introduce competing cations ($K^+$) and complexing anions ($Cl^-$), which can reduce $Cd^{2+}$ adsorption [34]. Inert phases like $SiO_2$-rich domains function mainly as a relatively inert matrix that dilutes the density of active adsorption sites. Accordingly, pre-washing (to remove soluble salts), acid/base treatments, or post-activation (e.g., HCl washing, EDTA extraction, chemical activation) are recommended to mitigate these inhibitory effects and further improve adsorption performance in future applications. Besides the (002) crystal plane of graphitic carbon, the (100) and (101) crystal planes are also observed, and the typical graphitized carbon diffraction peaks prove that TB and HB possess the fundamental properties of carbon materials. Moreover, this correlates with higher pyrolysis temperatures and degree

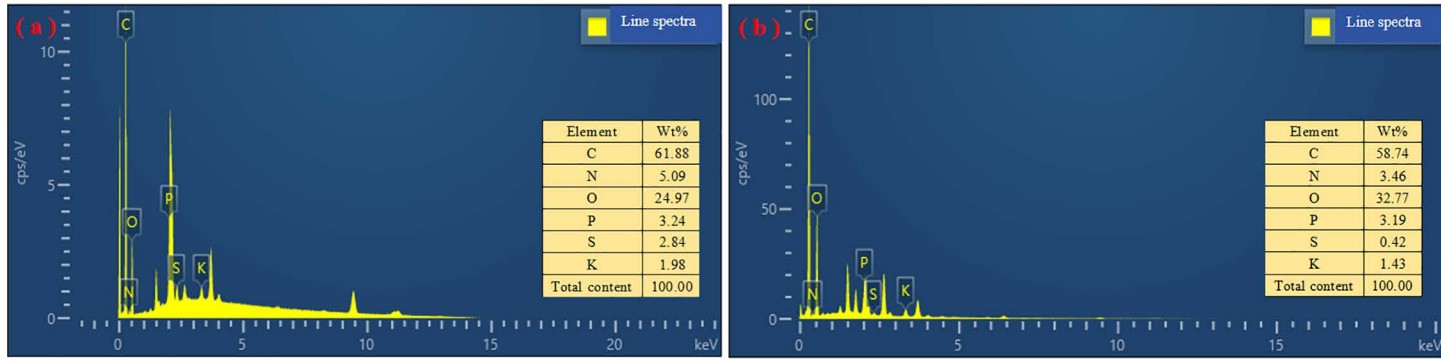

**Fig 2. EDS surface elemental analysis of the prepared materials. (a)** TB; **(b)** HB.

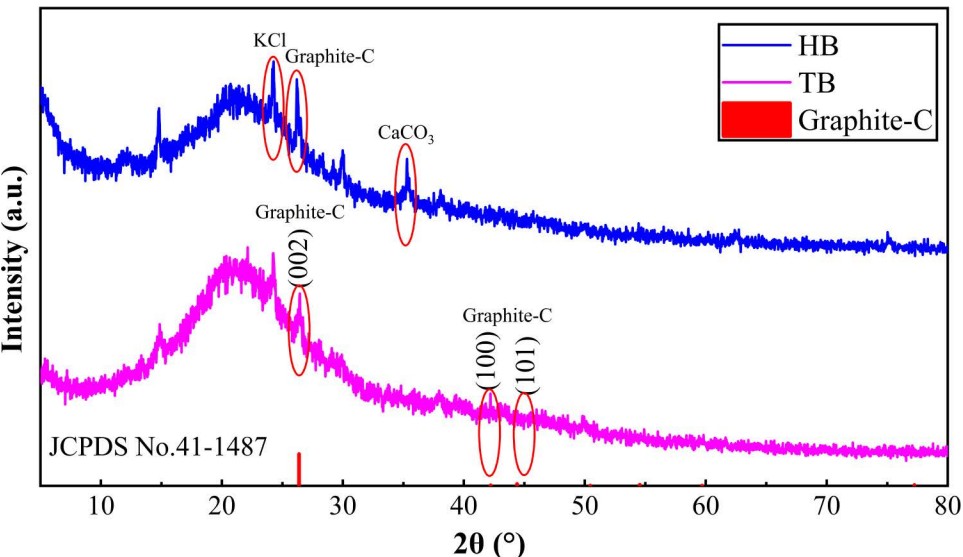

**Fig 3. XRD patterns of the prepared materials.**

of graphitization. It has also been suggested that graphitic carbon is an important basis for the ability of biochar to be used as a catalyst or adsorbent, which plays a significant role in environmental applications [35].

**FTIR analysis.** The FTIR spectra of the prepared materials are presented in Fig 4. It can be clearly seen that TB and HB have about 6 identical peak positions, which are located near 500−900, 1030, 1395, 1612, 2920, and 3426 $cm^{-1}$. Searching the spectrum data table [36], it was found that biochar has —OH vibration absorption peak (3426 $cm^{-1}$), and the C—H in alkanes were mainly vibration absorption peaks of methyl and methylene groups (2920 $cm^{-1}$), the aromatic acids —COOH groups (1612 $cm^{-1}$), fatty C—H and —C＝O groups (1395 $cm^{-1}$), C—C or C＝C resonance peaks (1030 $cm^{-1}$), and the pyridine and heterocyclic functional groups of aromatic compounds (500−900 $cm^{-1}$). It shows that TB and HB have rich functional groups on their surface. Qualitatively, the position and curve shape of each peak were generally similar, indicating that the functional group types of TB and HB were basically the same. However, there is some variation in the concentration of these functional groups. Specifically, TB has a higher concentration of —OH (3426 $cm^{-1}$) and heterocyclic functional groups of aromatic compounds (near 500−900 $cm^{-1}$) than HB, and a lower concentration of aromatic acids —

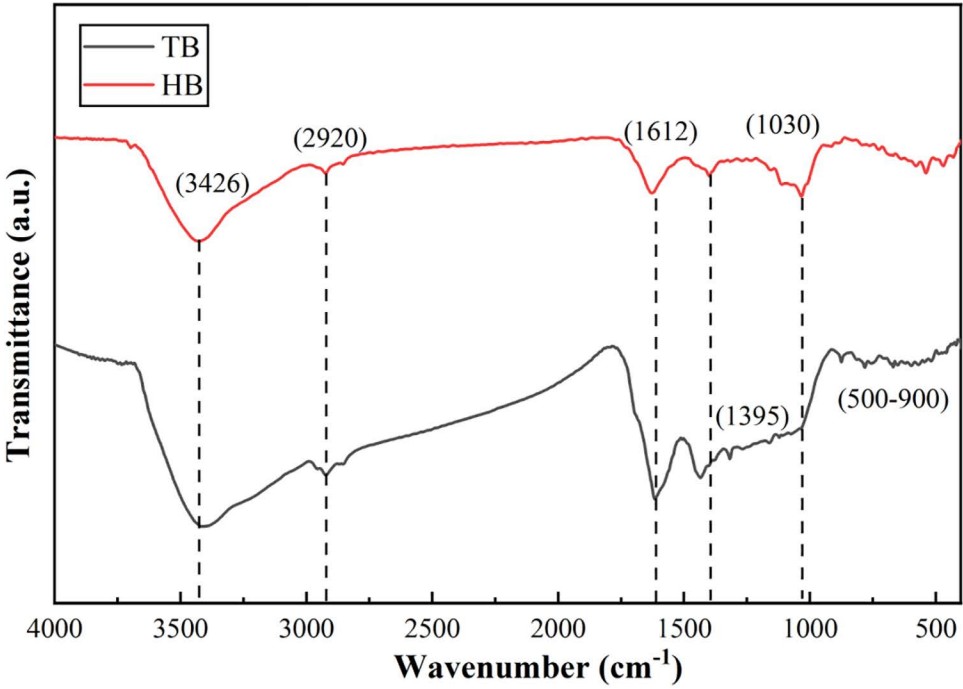

**Fig 4. FTIR spectra of the prepared materials.**

COOH groups (near 1612 cm$^{-1}$) and C—C or C=C groups (1030 cm$^{-1}$) than HB. However, this does not infer the contents of functional groups on the surface of TB and HB which need to be further explored.

**Specific surface area and pore size analysis.** The specific surface area and pore properties of the prepared materials are shown in Table 2. The adsorption-desorption isotherm of the prepared materials is shown in Fig 5. According to the definition of the International Union of Pure and Applied Chemistry (IUPAC) [37], the pore structure of TB and HB is mesoporous with average pore sizes ranging from 2 to 50 nm in this study. As could be seen from Fig 5a, the isothermal adsorption-desorption curve of TB is not closed, which may be due to its low pore size and very small specific surface area, as confirmed by the data in Table 2. Fig 5b shows that the isothermal adsorption-desorption curve of HB is basically IV(a) and there is a back hysteresis loop [38,39]. Quantitatively, the BET specific surface area of HB (1.867 m$^2$·g$^{-1}$) and total pore volume (0.003496 cm$^3$·g$^{-1}$) were markedly higher than those of TB (0.416 m$^2$·g$^{-1}$ and 0.000051 cm$^3$·g$^{-1}$, respectively), as shown in Table 2 and Fig 5, consistent with the more developed porous morphology observed in FESEM images. However, the specific surface area of HB is still small relative to other biochar prepared from biomass [31], which is highly related to its mixture of raw materials. To enhance the potential of HB for environmental applications, it should be further activated and treated.

## Bio-oil analysis

The main components of bio-oil produced by the pyrolysis process of tubers and herbal biomass are shown in Tables 3,4, respectively, and the total ion flow plots of PYGC-MS correspond to Fig 6(a),(c). In general, the chromatographic peak positions and chemical compositions of the bio-oils produced from tubers and herbal biomass were very similar in characteristics. After reviewing relevant information and searching in standard mass spectral libraries [23,40–43], it can be found that a total of 120 chromatographic peaks were identified in bio-oils produced from tubers biomass and 105 peaks were identified in bio-oils produced from herbal biomass. This means that the bio-oil produced during the high-temperature

**Table 2. The surface characteristics of TB and HB.**

| Surface feature | Test item | Unit | Type TB | Type HB | Remarks |
|---|---|---|---|---|---|
| Surface area | Single point surface area | m²·g⁻¹ | 0.168 | 1.177 | Relative pressure ($P/P_0$)=0.185; Where $P$ is the adsorption pressure and $P_0$ is the saturated vapor pressure of the adsorbate |
| | BET Surface Area | | 0.416 | 1.867 | Data fetching range 0.055~0.201 |
| | Langmuir Surface Area | | 3.619 | 8.232 | Monolayer adsorption model calculation |
| | BJH Adsorption cumulative surface area | | 0.113 | 1.102 | The aperture range is 2~55 |
| | t-Plot Micropore Area | | 0.988 | 0.788 | The aperture range is 2~55 |
| Pore volume | Single point adsorption total pore volume | cm³·g⁻¹ | 0.000051 | 0.003496 | When $P/P_0$=0.95, the total pore volume smaller than the critical pore diameter 55 |
| | BJH Adsorption cumulative volume | | 0.001947 | 0.011342 | The aperture range is 2~55 |
| | t-Plot micropore volume | | 0.000270 | 0.000301 | The aperture range is 2~55 |
| Pore size | Adsorption average pore diameter | nm | 2.487 | 7.490 | Calculated from $4V/A$, where the A corresponds to the adsorption BET specific surface area, adsorption cumulative pore internal surface area, desorption cumulative pore internal surface area, respectively; $V$ is the gas adsorption volume |
| | Desorption average pore diameter | | 4.914 | 8.980 | |
| | BJH Adsorption average pore width | | 49.146 | 41.173 | |

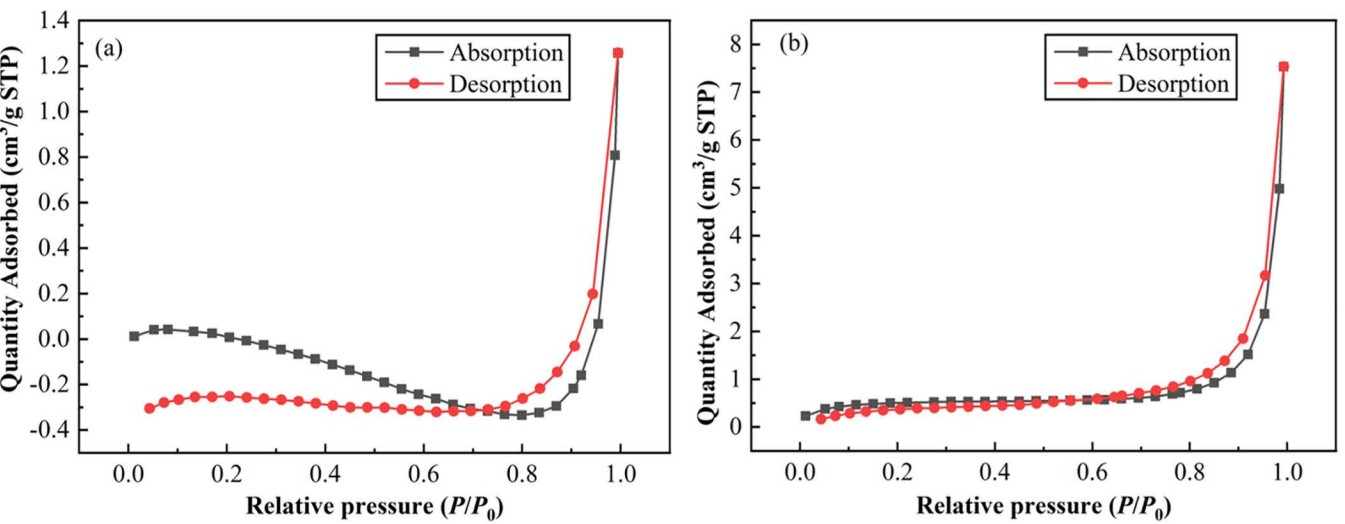

**Fig 5. Adsorption-desorption isotherm of TB (a) and HB (b).**

pyrolysis of pharmaceutical residues biomass may consist of more than 100 chemical substances. According to the classification of the main organic compounds in Tables 3,4, it could be found that some aliphatic compounds have relatively high content, including ketones, alkanes, alcohols, olefins, aldehydes, fatty acids, and so on. This is followed by aromatic compounds, including phenol, p-phenol, and their derivatives, etc. In addition, there are a variety of heterocyclic

**Table 3. The main components of bio-oil analyzed by PYGC-MS from tubers.**

| Peak number | Library/ID | CAS No. | Chemical formula | Ret. time (min) | Peak area (%) |
|---|---|---|---|---|---|
| 1 | Ethanol, 2,2-diethoxy- | 000621-63-6 | $C_6H_{14}O_3$ | 4.483 | 7.25 |
| 2 | Phenol | 000108-95-2 | $C_6H_6O$ | 5.334 | 6.38 |
| 3 | Butanoic acid, 2-methylphenyl ester | 014617-99-6 | $C_{11}H_{14}O_2$ | 6.735 | 5.61 |
| 4 | Propanoic acid, 2-hydroxy-, ethyl ester, (L)- | 000687-47-8 | $C_5H_{10}O_3$ | 3.051 | 3.49 |
| 5 | 2-Cyclopenten-1-one | 000930-30-3 | $C_5H_6O$ | 3.283 | 3.33 |
| 6 | 1,2-Cyclopentanedione, 3-methyl- | 000765-70-8 | $C_6H_8O_2$ | 5.956 | 3.02 |
| 7 | 2-Furaldehyde diethyl acetal | 013529-27-6 | $C_9H_{14}O_3$ | 6.711 | 2.58 |
| 8 | Furan, 2,5-diethoxytetrahydro- | 003320-90-9 | $C_8H_{16}O_3$ | 5.769 | 2.39 |
| 9 | Pentanoic acid, 4-oxo-, ethyl ester | 000539-88-8 | $C_7H_{12}O_3$ | 6.476 | 2.24 |
| 10 | Phenol, 3-methyl- | 000108-39-4 | $C_7H_8O$ | 6.412 | 2.07 |
| 11 | 2-Cyclopenten-1-one, 3-methyl- | 002758-18-1 | $C_6H_8O$ | 5.031 | 2.00 |
| 12 | 2-Pentanone, 5,5-diethoxy- | 014499-41-3 | $C_9H_{18}O_3$ | 8.000 | 1.94 |
| 13 | 1,4:3,6-Dianhydro-α-d-glucopyranose | -ᵃ | $C_6H_8O_4$ | 9.204 | 1.48 |
| 14 | Heptane, 1,1-diethoxy- | 000688-82-4 | $C_{11}H_{24}O_2$ | 12.343 | 1.39 |
| 15 | 2-Furancarboxaldehyde, 5-methyl- | 000620-02-0 | $C_6H_6O_2$ | 5.010 | 1.39 |
| 16 | 2-Cyclopenten-1-one, 2,3-dimethyl- | 001121-05-7 | $C_7H_{10}O$ | 6.133 | 1.37 |
| 17 | 4-Isopropyl-5,5-dimethyl-5H-furan-2-one | 060845-38-7 | $C_9H_{14}O_2$ | 8.228 | 1.34 |
| 18 | 2-Azaquinuclidone-3 | – | $C_6H_{10}N_2O$ | 8.065 | 1.14 |
| 19 | 2-Cyclopenten-1-one, 2-methyl- | 001120-73-6 | $C_6H_8O$ | 4.198 | 1.09 |
| 20 | 2(5H)-Furanone | 000497-23-4 | $C_4H_4O_2$ | 4.364 | 1.01 |
| 21 | Xanthine, 1,3-dipropyl-8-[4-[β-[(benzyloxy carbonylamino)acetylamino] | 102255-67-4 | $C_{31}H_{37}N_7O_7$ | 6.412 | 2.07 |

ᵃThis means the default CAS data.

compounds that are also important components of bio-oils, mainly involving pyridine, furan, pyrrole, and thiophene. In general, the content of aliphatic compounds in bio-oils is higher than that of aromatic compounds, with a ratio of about 2.8:1. Based on the data in Table 3, it could be found that the highest concentration of bio-oil produced by the tubers was "Ethanol, 2,2-diethoxy-", with a percentage of about 7.25%, which could be used as a fuel. This was followed by Phenol with a percentage of about 6.38%. In addition, the higher concentrations were Butanoic acid and Propanoic acid. α-d-glucopyranose and Xanthine have also been found, which can be used as high-value chemicals, which are valuable for pharmaceutical research. Bio-oils also contain a variety of ketones, benzenes, and heterocyclic compounds that could be used as important chemical raw materials, such as intermediates in the synthesis of drugs, dyes or fragrances [44]. Furthermore, it has also been suggested that if the bio-oil contains more than 20% of gasoline and diesel when the content of alkanes, especially C6-C15 hydrocarbons, is high, it means that the bio-oil extracted from biomass may be used as a potential biodiesel [23,45].

According to the data in Table 4, the highest concentration of bio-oil produced by the herbal was "Phosphonic acid, (p-hydroxyphenyl)- " with about 10.52%, which can be used as a raw material for compound fertilizer. This was followed by "Phenol, 3-methyl-" and "Ethanol, 2,2-diethoxy-" with 3.03% and 2.53% respectively. This is similar to the composition of bio-oil produced from tubers biomass. Moreover, bio-oils produced from both tubers and herbal biomass contain significant amounts of ketones and heterocyclic compounds, with a total content of more than 19%, which could be used as important chemical feedstocks. Comparison of Tables 3,4 reveals that the bio-oil produced from tubers biomass is more

**Table 4. The main components of bio-oil analyzed by PYGC-MS from herbal.**

| Peak number | Library/ID | CAS No. | Chemical formula | Ret. time (min) | Peak area (%) |
|---|---|---|---|---|---|
| 1 | Phosphonic acid, (p-hydroxyphenyl)- | 033795-18-5 | $C_6H_7O_4P$ | 5.351 | 10.52 |
| 2 | Phenol, 3-methyl- | 000108-39-4 | $C_7H_8O$ | 6.738 | 3.03 |
| 3 | Ethanol, 2,2-diethoxy- | 000621-63-6 | $C_6H_{14}O_3$ | 4.528 | 2.53 |
| 4 | Phenol, 4-ethyl- | 000123-07-9 | $C_8H_{10}O$ | 8.282 | 1.73 |
| 5 | trans-2-Methyl-4-hexen-3-ol | 096346-76-8 | $C_7H_{14}O$ | 6.769 | 1.34 |
| 6 | 4-Dehydroxy-N-(4,5-methylenedioxy-2-nitrobenzylidene) tyramine | -ᵃ | $C_{16}H_{14}N_2O_4$ | 3.983 | 1.26 |
| 7 | Furan, 2,5-diethoxytetrahydro- | 003320-90-9 | $C_8H_{16}O_3$ | 5.789 | 1.21 |
| 8 | Phenol, 3-methyl- | 000108-39-4 | $C_7H_8O$ | 6.432 | 0.95 |
| 9 | 3,3-Diethoxy-1-propanol, propyl ether | – | $C_{10}H_{22}O_3$ | 7.102 | 0.93 |
| 10 | Phenol, 3-chloro- | 000108-43-0 | $C_6H_5ClO$ | 9.061 | 0.76 |
| 11 | Pentanoic acid, 4-oxo-, ethyl ester | 000539-88-8 | $C_7H_{12}O_3$ | 6.504 | 0.75 |
| 12 | 2-Cyclopenten-1-one, 3-methyl- | 002758-18-1 | $C_6H_8O$ | 5.096 | 0.71 |
| 13 | 2-Pentanone, 5,5-diethoxy- | 014499-41-3 | $C_9H_{18}O_3$ | 8.021 | 0.67 |
| 14 | p-Cresol | 000106-44-5 | $C_7H_8O$ | 6.738 | 3.03 |

ᵃThis means the default CAS data.

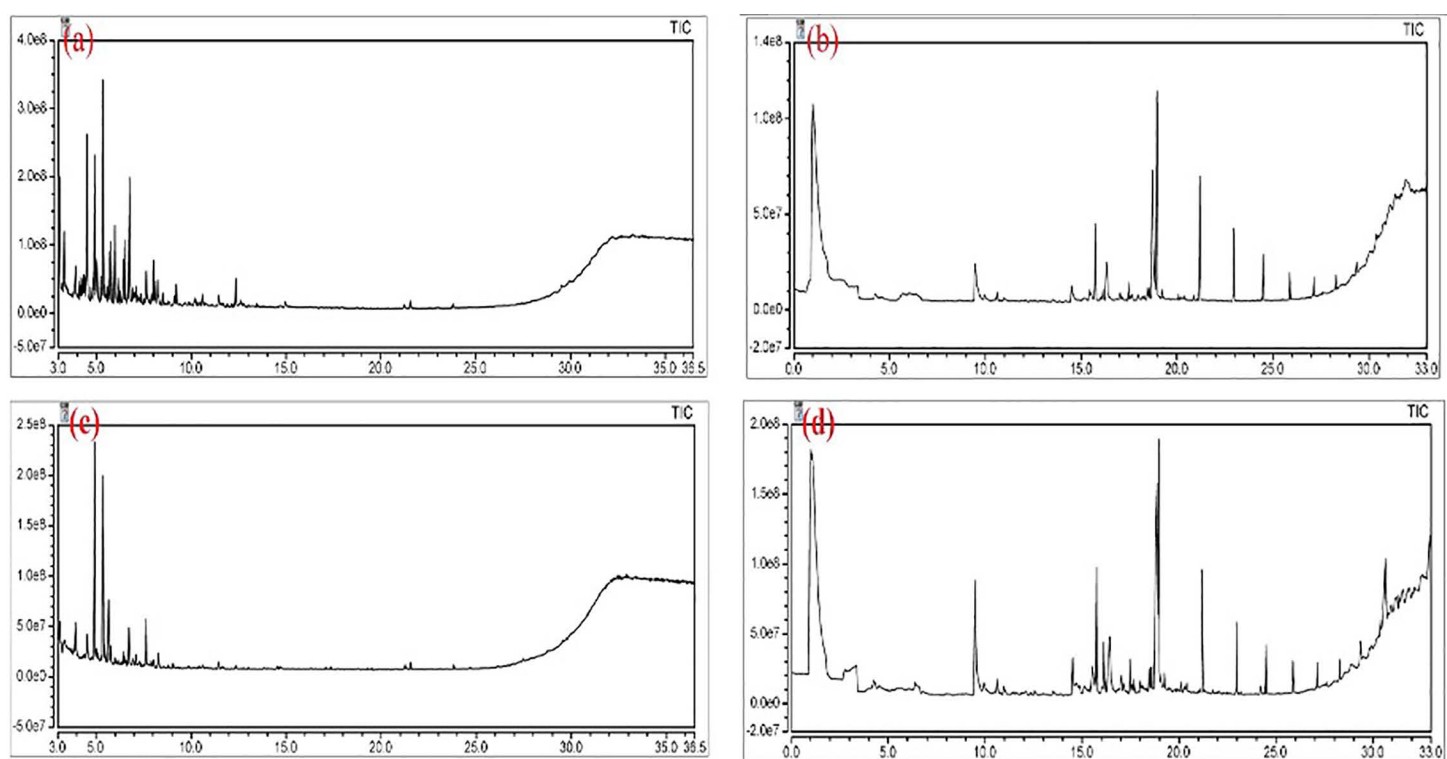

**Fig 6. Total Ion Chromatography. (a)** Bio-oil from tubers, **(b)** Syngas from tubers, **(c)** Bio-oil from herbal, and **(d)** Syngas from herbal.

chemically complex and less pure. Whereas the concentration of "Phosphonic acid, (p-hydroxyphenyl)-" in bio-oil produced from herbal biomass is significantly higher. If the possibility of practical utilization is taken into account, the bio-oil produced from herbal biomass is more advantageous.

## Syngas analysis

The main components of syngas produced by the pyrolysis process of tubers and herbal biomass are shown in Tables 5,6, respectively, and the total ion flow plots of PYGC-MS correspond to Figs 6(b),(d). Based on the results of Figs 6(b),(d), it was also observed that the syngas generated from the pyrolysis process of tubers and herbaceous biomass had basically similar chromatographic peak positions and chemical compositions, which were attributed to their similar feedstocks. According to the above bio-oil analysis results, it could be found that a total of 189 chromatographic peaks were identified in the syngas produced from tubers and herbal biomass, which also contained a wide range of aliphatic compounds, aromatic compounds, and heterocyclic compounds, with the highest percentage of aliphatic compounds, including alkanes, amino derivatives, alcohols, and derivatives. From Table 5, it can be observed that the highest concentration of syngas produced from tuberous biomass is Carbon dioxide with a percentage of about 8.38%, which is attributed to the products of biomass combustion or derivatives of carbon monoxide. It was followed by "Carbamic acid, monoammonium salt," "Hydrazinecarboxamide," "β-D-glucopyranose, 1,6-anhydro-," "4-Pyridinol," "3,4-Altrosan," and "D-Allose," with 7.40%, 6.05%, 4.99%, 4.82%, 4.42%, and 3.38%, respectively. From Table 6, it could be observed that the highest concentration of syngas produced from herbaceous biomass was D-Allose with about 8.46%, followed by "Carbamic acid, monoammonium salt," "3-Pyridinol," and "Carbon dioxide" with 8.38%, 4.82%, and 4.34%, respectively. As a result, the

**Table 5. The main components of syngas analyzed by PYGC-MS from tubers.**

| Peak number | Library/ID | CAS No. | Chemical formula | Ret. time (min) | Peak area (%) |
|---|---|---|---|---|---|
| 1 | Carbon dioxide | 000124-38-9 | $CO_2$ | 0.959 | 8.38 |
| 2 | Carbamic acid, monoammonium salt | 001111-78-0 | $CH_6N_2O_2$ | 1.132 | 7.40 |
| 3 | Hydrazinecarboxamide | 000057-56-7 | $CH_5N_3O$ | 1.282 | 6.05 |
| 4 | β-D-Glucopyranose, 1,6-anhydro- | 000498-07-7 | $C_6H_{10}O_5$ | 18.818 | 4.99 |
| 5 | 4-Pyridinol | 000626-64-2 | $C_5H_5NO$ | 9.492 | 4.82 |
| 6 | 3,4-Altrosan | -[a] | $C_6H_{10}O_5$ | 18.856 | 4.42 |
| 7 | D-Allose | 002595-97-3 | $C_6H_{12}O_6$ | 16.431 | 3.38 |
| 8 | Propanamide, N-methyl-2-amino- | 070875-70-6 | $C_4H_{10}N_2O$ | 1.439 | 2.13 |
| 9 | 1-(2-Aminopropoxy)-2-methoxyethane | 1038338-13-4 | $C_6H_{15}NO_2$ | 1.541 | 1.89 |
| 10 | Ethyne, fluoro- | 02713-09-9 | $C_2HF$ | 1.187 | 1.83 |
| 11 | Propanamide, N-methyl-2-amino- | 070875-70-6 | $C_4H_{10}N_2O$ | 1.496 | 1.48 |
| 12 | 2-Butene-1,4-diol, (Z)- | 006117-80-2 | $C_4H_8O_2$ | 15.539 | 1.18 |
| 13 | 2-Propanamine | 000075-31-0 | $C_3H_9N$ | 1.364 | 1.09 |
| 14 | Hydrazinecarboxamide | 000057-56-7 | $CH_5N_3O$ | 1.262 | 1.08 |
| 15 | Ribitol | 000488-81-3 | $C_5H_{12}O_5$ | 16.131 | 1.04 |
| 16 | 2,5-Furandione, 3-dodecyl- | 059426-46-9 | $C_{16}H_{26}O_3$ | 30.936 | 1.03 |
| 17 | Phenol, 2,6-bis(1,1-dimethylethyl)- | 000128-39-2 | $C_{14}H_{22}O$ | 30.654 | 0.95 |
| 18 | 17-Pentatriacontene | 006971-40-0 | $C_{35}H_{70}$ | 31.222 | 0.83 |
| 19 | Phloroglucitol | 002041-15-8 | $C_6H_{12}O_3$ | 14.679 | 0.80 |
| 20 | 3-(4-(tert-Butyl)phenyl)-2-methylpropan-1-ol | 056107-04-1 | $C_{14}H_{22}O$ | 30.511 | 0.75 |

[a]This means the default CAS data.

**Table 6. The main components of syngas analyzed by PYGC-MS from herbal.**

| Peak number | Library/ID | CAS No. | Chemical formula | Ret. time (min) | Peak area (%) |
|---|---|---|---|---|---|
| 1 | D-Allose | 002595-97-3 | $C_6H_{12}O_6$ | 18.856 | 8.46 |
| 2 | Carbamic acid, monoammonium salt | 001111-78-0 | $CH_6N_2O_2$ | 1.024 | 8.38 |
| 3 | 3-Pyridinol | 000109-00-2 | $C_5H_5NO$ | 9.492 | 4.82 |
| 4 | Carbon dioxide | 124-38-9 | $CO_2$ | 1.132 | 4.34 |
| 5 | 1,6-Anhydro-β-d-talopyranose | -[a] | $C_6H_{10}O_5$ | 16.431 | 2.36 |
| 6 | Carbamic acid, monoammonium salt | 001111-78-0 | $CH_6N_2O_2$ | 0.959 | 2.15 |
| 7 | Ethyne, fluoro- | 002713-09-9 | $C_2HF$ | 1.439 | 2.13 |
| 8 | 3,5-Decadien-7-yne, 6-t-butyl-2,2,9,9-tetramethyl- | – | $C_{18}H_{30}$ | 30.627 | 1.35 |
| 9 | Amberonne (isomer 1) | – | $C_{16}H_{26}O$ | 30.599 | 1.19 |
| 10 | β-D-Glucopyranoside, methyl 3,6-anhydro- | 003056-46-0 | $C_7H_{12}O_5$ | 15.539 | 1.18 |
| 11 | 3-O-Methyl-d-glucose | – | $C_7H_{14}O_6$ | 16.131 | 1.04 |
| 12 | 3,4-Altrosan | – | $C_6H_{10}O_5$ | 18.835 | 1.02 |
| 13 | Hydroquinone | 000123-31-9 | $C_6H_6O_2$ | 14.512 | 0.95 |
| 14 | 1-Penten-3-one, 1-(2,6,6-trimethyl-1-cyclohexen-1-yl)- | 000127-43-5 | $C_{14}H_{22}O$ | 32.899 | 0.66 |
| 15 | tert-Hexadecanethiol | 025360-09-2 | $C_{16}H_{34}S$ | 30.817 | 0.51 |
| 16 | α-D-Glucopyranose, 4-O-β-D-galactopyranosyl- | 014641-93-1 | $C_{12}H_{22}O_{11}$ | 17.036 | 0.43 |
| 17 | Phenol, 2,6-bis(1,1-dimethylethyl)- | 000128-39-2 | $C_{14}H_{22}O$ | 32.922 | 0.43 |
| 18 | 1,3-Cyclopentanediol, trans- | 016326-98-0 | $C_5H_{10}O_2$ | 15.114 | 0.42 |
| 19 | 3-Pyridinol, 2-methyl- | 001121-25-1 | $C_6H_7NO$ | 9.934 | 0.42 |
| 20 | 3-Methyldodecanoic acid | 013490-36-3 | $C_{13}H_{26}O_2$ | 19.253 | 0.36 |

[a]This means the default CAS data.

syngas from the two biomasses contain essentially no reducing hydrogen content [46], but instead more exhaust from pyrolysis, which has a limited number of chemicals available, mostly amino derivatives and alcohol derivatives, in low concentrations. Comparing the data in Tables 5,6, the concentration of exhaust gases, such as $CO_2$, in syngas produced from herbaceous biomass is lower than in syngas produced from tuberous biomass, and the availability of herbaceous biomass is relatively more promising [47].

## Discussions

**Pyrolysis mechanisms.** Typically, biomass is composed of complex polymeric organic polymers such as cellulose, hemicellulose, and lignin, and biomass pyrolysis is a combination process of these three main components. According to Aykac et al. [42], it can be found that the biomass contains 35–43% of holocellulose, of which a-cellulose accounts for 21–29%. Unfortunately, there are limited studies and referenceable literature on the pyrolysis mechanism, and based on the results of Fakayode et al. [48] and Wang et al. [23], we assumed that the possible pyrolysis processes of the biomass consist of three stages. In the first stage, the water in the biomass is evaporated at 105°C. With the increase in temperature into the second stage, at this time holocellulose began to pyrolyze, mainly in the hemicellulose at 200–295 °C softening and cellulose at 240–375°C decomposition, the products generated mainly for the volatile matters and a few biochar. The volatile matter consists of condensable and non-condensable gases, with the condensable gas condensing rapidly to obtain bio-oil and the non-condensable gas being syngas [28]. The volatile matter is mostly acidic, and some acidic or oxygenated functional groups are lost at this stage. In addition, the biochar walls become thinner, the diameter of the fibers decreases, and the surface shows a small amount of porosity, which is associated with a rapid accumulation and sustained release of syngas. In the third stage at 280–500°C, lignin decomposition produces mainly biochar and

some volatile matters. The pore structure of biochar tended to become more complex and homogeneous with the rapid aggregation and precipitation of volatile fractions and the increasing number of gas products, and the simple pore structure became dense and diverse, forming a large number of mesopores around and inside the larger pores. The alkaline functional groups and aromaticity of biochar were enhanced at this stage [49]. The continuous increase of volatile matter and syngas has made the pore structure of biochar more complicated and uniform. It is reflected in the simple pore structure becoming dense and diverse, and a large number of mesopores are formed around and inside the larger pores. Moreover, the alkaline functional groups and the aromaticity of biochar are enhanced at this stage. In addition, the compositions of the bio-oil obtained by Zhang *et al*. [28] during the hydrothermal process had many similarities with the compositions analyzed by GC-MS in this work, especially the presence of high quality chemicals dimethoxyphenol (syringol) and 2-methoxyphenol (guaiacol) in both. However, the compositions of the bio-oil from pyrolysis are more complex and contain more organic compounds than that from hydrothermal carbonization. Finally, we observed a change in syngas color from light to dark to lighter during pyrolysis, and the bio-oil and syngas production also showed an increase first and then decrease with increasing pyrolysis temperature.

**Properties analysis of different biomass pyrolysis products.** In this work, the bio-oil produced from the pyrolysis of both tubers and herbal biomass contained a high concentration of aliphatic compounds, which could be used as a potential biodiesel, attributed to its high level of hydrocarbons. Bio-oils also contain a certain quantity of high-value chemicals, such as guaiacol (phenol, 2-methoxy-) and α-D-glucose (1,4:3,6-dihydro-α-D-glucopyranose), which are aromatic or heterocyclic compounds that can be utilized as important chemical raw materials. Guaiacol (The bio-oil related content from tubers and herbals was 2.07% and 3.03%, respectively) is an important fine chemical intermediate, widely used in the synthesis of drugs, spices, and dyes. α-D-glucopyranose (relative content 1.48%) is rarely synthesized considering its cost and is mainly used in medical liver function tests or pharmaceuticals. The presence of α-D-glucopyranose in bio-oil is an interesting discovery to explore. Thus, bio-oil is a valuable product obtained from pyrolyzed biomass, usually in the form of a dark brown organic liquid mixture. A comparison of some relevant literature reports [23, 50] reveals that in woody biomass, all bio-oils produced from pyrolysis contain guaiacol and α-D-glucose. Wang *et al*. [23] reported the chemical composition of bio-oil produced from *Caragana korshinskii* as an agroforestry waste, and he concluded that biomass bio-oil has the advantages of a wide range of feedstock sources, abundant and renewable reserves, high energy density, and ease of transportation and storage. Li *et al*. [41] and Khuenkaeo *et al*. [49] also considered bio-oil as a potential source of liquid fuels and chemical feedstocks. Biomass pyrolysis is an important technological approach to solve the fossil energy problem in the future, and many scholars have reached a consensus on this point [50]. Of course, to better utilize these bio-oils there is a need to continue the research related to hydrocracking or catalytic cracking in order to enhance the purity of biodiesel [43,45].

Bio-oil also contains a variety of ketones, benzenes, and heterocyclic compounds, which act as important chemical raw materials, such as Xanthine, which can be used as intermediates for the synthesis of drugs, dyes, and fragrances, and are very valuable for pharmaceutical research. However, the syngas from pyrolysis of tubers or herbal biomass is essentially devoid of carbon monoxide and hydrogen, and most of its available chemicals are amino and alcohol derivatives, which means that applications are of little value and that the rate of pyrolysis needs to be varied with a view to producing more solids and liquids and less off-gas from pyrolysis of pharmaceutical residues. After comparison, it was found that herbaceous biomass has a higher availability value, which is related to the higher content of cellulose and hemicellulose in its chemical composition, and may also be closely related to the greater susceptibility to pyrolysis processes [23, 47].

### Environmental applications

**Isothermal adsorption analysis.** As shown in the isothermal adsorption curve of $Cd^{2+}$ by the prepared materials in Fig 7, both TB and HB exhibit a gradual increase in $Cd^{2+}$ adsorption capacity with rising equilibrium concentrations, reaching saturation in the high-concentration range and demonstrating typical monolayer adsorption characteristics.

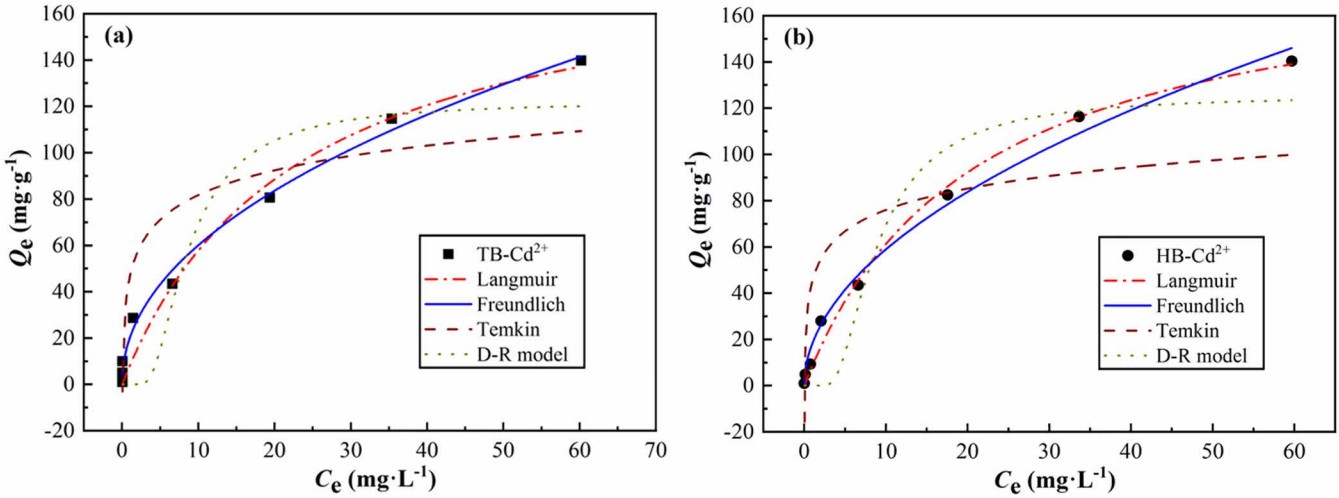

**Fig 7. Isotherm adsorption curves of Cd2+ by TB and HB. (a)** for TB, **(b)** for HB.

When the initial concentration increased from 1 mg·L⁻¹ to 250 mg·L⁻¹, the equilibrium adsorption capacity of TB rose from 0.96 mg·g⁻¹ to 139.79 mg·g⁻¹, while that of HB increased from 0.99 mg·g⁻¹ to 140.33 mg·g⁻¹. This indicates that both biochar types retain substantial Cd²⁺ loading capacity under high-concentration conditions. The adsorption capacities of both materials showed minimal differences in the low-concentration range (1–10 mg·L⁻¹), while exhibiting nearly identical adsorption capacities in the medium-to-high concentration range (50–250 mg·L⁻¹). HB demonstrated slightly higher adsorption capacity in the low-concentration range, likely attributed to its larger specific surface area and more developed pore structure, consistent with the previous characterization results. Overall, the Cd²⁺ adsorption behavior of both biochar types aligns with typical adsorbent characteristics, demonstrating potential for further application in heavy metal remediation of water bodies.

To further investigate the adsorption mechanism, this study employed four models—Langmuir, Freundlich, Temkin, and Dubinin-Radushkevich—to fit the adsorption data. As shown in Table 7, the Freundlich model fitted coefficients ($R^2$) for both biochar types exceeded 0.99, indicating that their Cd²⁺ adsorption better aligns with a multi-layer heterogeneous adsorption process. The Langmuir model also demonstrated good fitting performance ($R^2 > 0.97$), yielding calculated maximum adsorption capacities of 188.89 mg·g⁻¹ (TB) and 186.67 mg·g⁻¹ (HB), further validating their high adsorption capacities. The Temkin and D-R models showed relatively poor fitting results, suggesting potential energy heterogeneity in the adsorption process, though physical adsorption and surface complexation remained the dominant mechanisms overall [51].

**Adsorption kinetic analysis.** Fig 8 displays the adsorption kinetic curves of TB and HB for Cd²⁺. It can be observed that both biochar types exhibit rapid adsorption followed by equilibrium for Cd²⁺. Within the first 300 min, the adsorption

**Table 7. Parameters for the isotherm adsorption model of Cd²⁺ by TB and HB.**

| Materials | Langmuir | | | Freundlich | | | Temkim | | | D-R model | | |
|---|---|---|---|---|---|---|---|---|---|---|---|---|
| | $a$ | $Q_m$/mg·g⁻¹ | $R^2$ | $K_F$ | $n$ | $R^2$ | $A$ | $B$ | $R^2$ | $Q_0$/mmol·g⁻¹ | $E$/kJ·mol⁻¹ | $R^2$ |
| TB-Cd²⁺ | 0.04 | 188.89 | 0.97 | 20.01 | 2.10 | 0.99 | 20.18 | 15.39 | 0.82 | 122.15 | 0.22 | 0.88 |
| HB-Cd²⁺ | 0.05 | 186.67 | 0.99 | 18.30 | 1.97 | 0.99 | 29.96 | 13.33 | 0.64 | 125.63 | 0.22 | 0.90 |

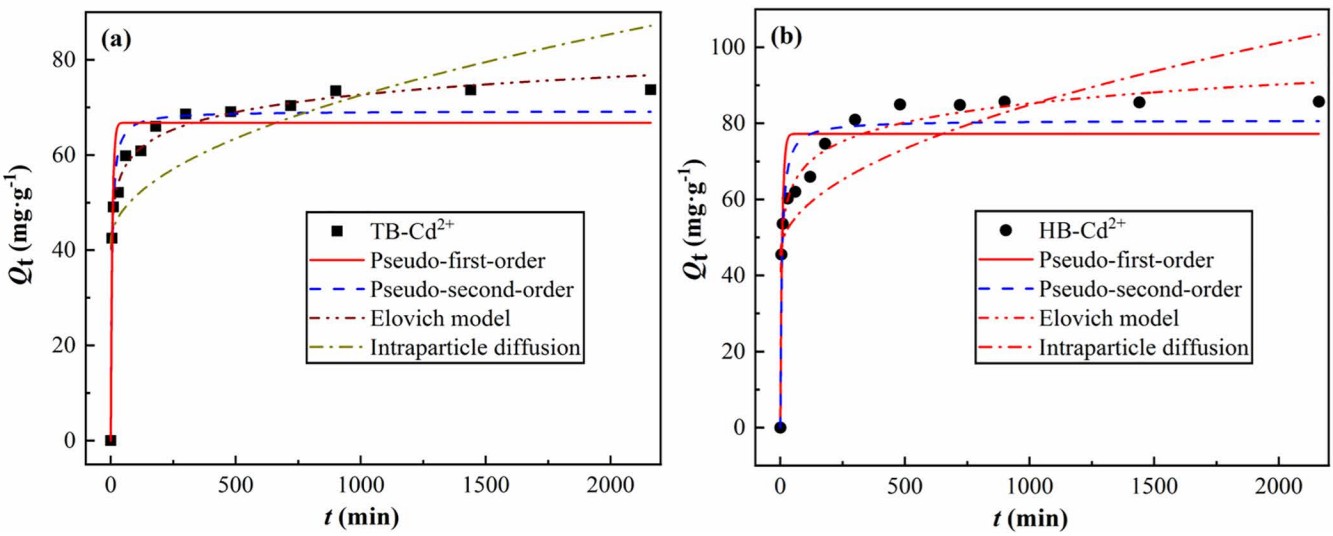

**Fig 8. Kinetic curves for the adsorption of Cd²⁺ by TB and HB. (a) for TB, (b) for HB.**

rate was relatively fast, with HB consistently showing higher adsorption capacity than TB, indicating superior initial adsorption performance. As time progressed, adsorption gradually reached equilibrium. At 2160 min, the adsorption capacities of TB and HB were 73.70 mg·g⁻¹ and 85.67 mg·g⁻¹, respectively, further confirming HB's superiority over TB in both adsorption capacity and rate. Moreover, HB exhibited slightly higher overall adsorption amounts than TB at all time points, suggesting that HB possesses faster or more effective adsorption sites/more favorable surface properties for Cd²⁺, resulting in superior equilibrium adsorption amounts and rates compared to TB. This kinetic behavior provides important guidance for designing contact time and treatment efficiency in practical wastewater treatment processes [23].

As shown in Table 8, the Elovich model provides the best fit for both biochars ($R^2 \geq 0.98$), indicating that the adsorption of Cd²⁺ on the biochar surface is a non-uniform diffusion process involving multiple active sites. The R² values of the pseudo-second-order kinetic model are also relatively high ($\geq 0.92$), suggesting that chemical adsorption plays a significant role in the process. In contrast, the pseudo-first-order model and the intra-particle diffusion model have poorer fitting effects, indicating that the adsorption process is not only controlled by diffusion but also influenced by surface reactions and pore structures, further confirming the characteristics of biochar as a composite adsorbent. Overall, the kinetic fitting results suggest that the adsorption process is a multi-mechanism composite control, with rapid surface adsorption/chemical adsorption at the initial stage, followed by limitations imposed by surface heterogeneity and intra-particle diffusion.

**Table 8. Parameters of the adsorption kinetic model of Cd²⁺ by TB and HB.**

| Materials | Pseudo-first-order | | | Pseudo-second-order | | | Elovich model | | | Intraparticle diffusion | | |
|---|---|---|---|---|---|---|---|---|---|---|---|---|
| | $Q_e$/mg·g⁻¹ | $k_1$ | $R^2$ | $Q_e$/mg·g⁻¹ | $k_2$ | $R^2$ | $a$ | $b$ | $R^2$ | $k_i$ | $c$ | $R^2$ |
| TB-Cd²⁺ | 66.76 | 0.37 | 0.89 | 63.22 | 0.003 | 0.95 | 86.45 | 0.19 | 0.99 | 0.98 | 41.54 | 0.45 |
| HB-Cd²⁺ | 77.26 | 0.32 | 0.84 | 80.79 | 0.002 | 0.92 | 98.00 | 0.12 | 0.98 | 1.25 | 45.46 | 0.52 |

## Conclusions

This study successfully demonstrated a sustainable strategy for the comprehensive utilization of waste pharmaceutical residues via oxygen-limited pyrolysis at 500 °C. Two types of biochar, TB and HB, were prepared and thoroughly characterized. The results confirmed that both biochars are mesoporous carbon materials enriched with oxygen-containing functional groups (e.g., hydroxyl, carboxyl, carbonyl) and graphitic carbon crystals, which are beneficial for environmental applications. Notably, HB, derived from herbal biomass, exhibited superior characteristics, including a larger specific surface area and pore volume, making it a more recommended carbon material. The parallel production of bio-oil and syngas was systematically evaluated. The bio-oils were rich in valuable chemicals, such as ketones, alkanes, alcohols, and heterocyclic compounds, identifying them as promising sources for liquid biofuels and chemical feedstocks. The highest abundance components were "Ethanol, 2,2-diethoxy-" (7.25%) in tuber-based and "Phosphonic acid, (p-hydroxyphenyl)-" (10.52%) in herbal-based bio-oil. While the syngas contained various organic compounds, its application potential was limited. Crucially, the environmental application of the biochars was validated through $Cd^{2+}$ adsorption experiments. Both TB and HB showed high adsorption capacities, with the process best described by the Freundlich isotherm ($R^2 \geq 0.99$) and the Elovich kinetic model ($R^2 \geq 0.98$), indicating multilayer adsorption on heterogeneous surfaces. HB consistently outperformed TB, achieving a higher equilibrium adsorption capacity (85.67 mg·g$^{-1}$) and faster kinetics, which is attributed to its more developed porous structure. Therefore, pyrolysis, particularly utilizing herbal biomass from pharmaceutical residues, presents an efficient "waste-to-wealth" approach, simultaneously generating an effective adsorbent for heavy metal remediation and high-quality bio-energy products.

Future research should focus on several promising avenues to bridge the gap between laboratory findings and practical implementation. Firstly, exploring various chemical or physical activation methods to further enhance the specific surface area and pore structure of the biochars could significantly boost their adsorption performance for a broader spectrum of pollutants, including other heavy metals and organic contaminants. Secondly, the catalytic upgrading of the crude bio-oil should be investigated to improve its quality and energy density, for instance, through catalytic pyrolysis or hydrodeoxygenation processes. Thirdly, techno-economic analysis and life cycle assessment are essential to evaluate the economic viability and environmental footprint of the entire pyrolysis valorization process at a pilot or industrial scale. Finally, testing the adsorption efficiency of these biochars in real industrial wastewater matrices, which contain complex compositions and competing ions, will be a critical step toward validating their practical application potential and understanding their behavior in realistic scenarios.

## Author contributions

**Conceptualization:** Tongtong Wang, Junchao Jia.

**Data curation:** Puyang Feng, Wenxiao Li, Tao Qin, Tongtong Wang, Xinjie Zhang.

**Formal analysis:** Zhizhen Feng.

**Funding acquisition:** Zhizhen Feng, Bo Fu, Shanshan Han, Tao Qin.

**Investigation:** Puyang Feng, Wenxiao Li, Tao Qin, Xinjie Zhang, Junchao Jia.

**Methodology:** Bo Fu, Shanshan Han.

**Project administration:** Junchao Jia.

**Resources:** Zhizhen Feng, Hong Yan, Junchao Jia.

**Supervision:** Zhizhen Feng, Bo Fu, Junchao Jia.

**Validation:** Hong Yan, Tongtong Wang.

**Writing – original draft:** Zhizhen Feng.

**Writing – review & editing:** Bo Fu, Shanshan Han.

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
