## [Decision Letter · Decision Letter 0]

12 Dec 2025

Dear Dr. Jia,

Thank you for submitting your manuscript to PLOS ONE. After careful consideration, we feel that it has merit but does not fully meet PLOS ONE’s publication criteria as it currently stands. Therefore, we invite you to submit a revised version of the manuscript that addresses the points raised during the review process.

We look forward to receiving your revised manuscript.

Kind regards,

Nor Adilla Rashidi, Ph.D.

Academic Editor

PLOS One

Journal Requirements:

“This research was supported by the Science and Technology Program of Shaanxi Academy of Sciences (Grant No.: 2024k-02), the Science and Technology Program of Xianyang (Grant No.: L2022-QCYZX-NY-006), the Shaanxi Provincial Key R&D Program (Grant No.: 2023‑ZDLNY‑54), and the Shaanxi Provincial Innovation Capacity Support Program (Grant No.: 2023-CX-PT-12).”

“This research was supported by the Science and Technology Program of Shaanxi Academy of Sciences (Grant No.: 2024k-02), the Science and Technology Program of Xianyang (Grant No.: L2022-QCYZX-NY-006), the Shaanxi Provincial Key R&D Program (Grant No.: 2023‑ZDLNY‑54), and the Shaanxi Provincial Innovation Capacity Support Program (Grant No.: 2023-CX-PT-12).”

“This research was supported by the Science and Technology Program of Shaanxi Academy of Sciences (Grant No.: 2024k-02), the Science and Technology Program of Xianyang (Grant No.: L2022-QCYZX-NY-006), the Shaanxi Provincial Key R&D Program (Grant No.: 2023‑ZDLNY‑54), and the Shaanxi Provincial Innovation Capacity Support Program (Grant No.: 2023-CX-PT-12).”

6. Please include a separate caption for each figure in your manuscript.

7. Please include your tables as part of your main manuscript and remove the individual files. Please note that supplementary tables (should remain/ be uploaded) as separate "supporting information" files.

Reviewers' comments:

Reviewer's Responses to Questions

**Comments to the Author**

1. Is the manuscript technically sound, and do the data support the conclusions?

Reviewer #1: Yes

Reviewer #2: Yes

2. Has the statistical analysis been performed appropriately and rigorously?

Reviewer #1: Yes

Reviewer #2: Yes

3. Have the authors made all data underlying the findings in their manuscript fully available?

Reviewer #1: Yes

Reviewer #2: Yes

4. Is the manuscript presented in an intelligible fashion and written in standard English?

Reviewer #1: Yes

Reviewer #2: Yes

**5. Review Comments to the Author**

Reviewer #1: Valorization of Waste Pharmaceutical Residues via Pyrolysis: Simultaneous production

of biochar for Cd2+ removal and high-quality bio-oil/syngas

The study has some deficiencies and can be accepted for publication after the corrections/additions listed below are completed:

1. The abstract provides a good overview, but some specific details could be enhanced for better clarity. For instance, mentioning the specific types of 'tuber' and 'herbal' pharmaceutical residues used (e.g., 'Changyanning Pill' and 'Xinsuning capsule' as noted in the materials section) would immediately provide context to readers unfamiliar with these terms. Additionally, while the abstract states that syngas showed 'limited application potential', a brief reason for this limitation would be beneficial, even if elaborated further in the main text.

2. Adsorption part was not discussed sufficiently in the introduction section of the study. I suggest that this section be improved with the following up-to-date references:

https://doi.org/10.1038/s41598-024-51587-6

https://doi.org/10.1007/s13201-023-02007-z

https://doi.org/10.5004/dwt.2022.28506

3. The introduction effectively highlights the problem of waste pharmaceutical residues in China and the need for high-value utilization. However, it could strengthen the justification for pyrolysis as the chosen method by briefly comparing its advantages over other mentioned methods (e.g., bioconversion, direct combustion) in the context of the specific waste materials studied.

4. In the 'Materials' section, the paper mentions that raw materials were collected from 'Shaanxi Momeide Pharmaceutical Co., LTD.'. While this is a good start, specifying the exact nature or form of these residues (e.g., pre-processed waste, raw plant material) would add important context for reproducibility.

5. The description of the pyrolysis process is clear regarding temperature, time, and atmosphere. However, mentioning the type of pyrolysis (e.g., slow pyrolysis, fast pyrolysis) more explicitly in the main text, beyond just 'oxygen-limited pyrolysis', would align with common terminology in the field.

6. The FESEM analysis mentions 'obvious honeycomb polyhedral structures' and 'abundant porous structures'. Including quantitative data or a clearer comparative statement about the pore size distribution (beyond just 'larger specific surface area and pore volume' for HB) would strengthen these observations.

7. For the XRD analysis, the presence of impurities like CaO, KCl, or SiO2 is mentioned. It would be helpful to discuss the potential impact of these impurities on the biochar's performance, especially for Cd2+ adsorption, or how they might be mitigated in future applications.

Reviewer #2: Valorization of Waste Pharmaceutical Residues via Pyrolysis: Simultaneous production of biochar for Cd2+ removal and high-quality bio-oil/syngas

This study focuses on production of bio-oil, syngas and biochar. The study has a robust contribution to knowledge as it demonstrates the application of biochar for heavy metal removal., however minor modifications are required to improve it.

1. In Section 2.1 the basic physicochemical properties of the two feedstock biomasses were mentioned but the procedure for the determination of the results in Table 1 was not given. Furthermore, the Table itself is an important part of the results and needs to be moved to the manuscript. The abstract needs to be modified to include the method too.

2. In line 68, page 4, the term pyrolysis gasification is misleading, while pyrolysis occur at the onset of gasification, it does not justify the usage in this context and several places in the manuscript. The authors may need to explain clearly.

3. Please reframe line 121, page six.

4. Please label Figure 1 for easy identification and explanation of the features.

5. The major peaks of Figures 3 and 6 needs to be labelled.

I therefore recommend minor revision before acceptance.

---

## [Author Response · Author response to Decision Letter 1]

27 Jan 2026

Dear Editors and Reviewers,

We wish to express our sincere gratitude for giving us the opportunity to revise our manuscript. We greatly appreciate the time and effort you and the reviewers have dedicated to providing constructive comments, which have significantly improved the quality of our work.

We have carefully reviewed the decision letter and the reviewers’ comments. The manuscript has been revised thoroughly in accordance with these suggestions and the editorial requirements. A detailed point-by-point response to each reviewer is provided in the following pages.

Regarding the Funding Statement, the correct statement reads as follows, please update the online submission form on our behalf.

“Funding Statement: This research was supported by the Science and Technology Program of Shaanxi Academy of Sciences (2024k-02), the Science and Technology Program of Xianyang (L2022-QCYZX-NY-006), the Province Key Research and Development Program of Shaanxi (2023-ZDLNY-54), and the Science and Technology Department of Shaanxi Province (Grant Nos. 2023-CX-PT-12 and 2025NC-YBXM-122). The funders mentioned in our study had no role in study design, data collection and analysis, decision to publish, or preparation of the manuscript.”

We hope that the editor and reviewers find our revisions satisfactory. If there are any remaining issues, please let us know, and we will address them promptly.

Thank you once again for your support and guidance throughout this process.

Yours sincerely,

Dr. Junchao Jia (on behalf of all authors)

---

## [Decision Letter · Decision Letter 1]

16 Mar 2026

Valorization of Waste Pharmaceutical Residues via Pyrolysis: Simultaneous production of biochar for Cd2+ removal and high-quality bio-oil/syngas

PONE-D-25-58122R1

Dear Dr. Jia,

We’re pleased to inform you that your manuscript has been judged scientifically suitable for publication and will be formally accepted for publication once it meets all outstanding technical requirements.

Kind regards,

Nor Adilla Rashidi, Ph.D.

Academic Editor

PLOS One

Reviewers' comments:

Reviewer's Responses to Questions

**Comments to the Author**

Reviewer #1: All comments have been addressed

2. Is the manuscript technically sound, and do the data support the conclusions?

Reviewer #1: Yes

3. Has the statistical analysis been performed appropriately and rigorously?

Reviewer #1: Yes

4. Have the authors made all data underlying the findings in their manuscript fully available?

Reviewer #1: Yes

5. Is the manuscript presented in an intelligible fashion and written in standard English?

Reviewer #1: Yes

---

## [Editor Report · Acceptance letter]

PONE-D-25-58122R1

PLOS One

Dear Dr. Jia,

I'm pleased to inform you that your manuscript has been deemed suitable for publication in PLOS One. Congratulations! Your manuscript is now being handed over to our production team.

Kind regards,

on behalf of

Dr. Nor Adilla Rashidi

Academic Editor

PLOS One